# Association between Vitamin D Serum Levels and Immune Response to the BNT162b2 Vaccine for SARS-CoV-2

**DOI:** 10.3390/biomedicines10081993

**Published:** 2022-08-17

**Authors:** Paola Zelini, Piera d’Angelo, Emanuele Cereda, Catherine Klersy, Peressini Sabrina, Riccardo Albertini, Giuseppina Grugnetti, Anna Maria Grugnetti, Carlo Marena, Sara Cutti, Daniele Lilleri, Irene Cassaniti, Baldanti Fausto, Riccardo Caccialanza

**Affiliations:** 1Molecular Virology Unit, Microbiology and Virology Department, Fondazione IRCCS Policlinico San Matteo, 27100 Pavia, Italy; 2Department of Obstetrics and Gynecology, Fondazione IRCCS Policlinico San Matteo, University of Pavia, 27100 Pavia, Italy; 3Clinical Nutrition and Dietetics Unit, Fondazione IRCCS Policlinico San Matteo, 27100 Pavia, Italy; 4Unit of Clinical Epidemiology and Biostatistics, Scientific Direction, Fondazione IRCCS Policlinico San Matteo, 27100 Pavia, Italy; 5Laboratory of Clinical Chemistry, Fondazione IRCCS Policlinico San Matteo, 27100 Pavia, Italy; 6Health Professions Direction, Fondazione IRCCS Policlinico San Matteo, 27100 Pavia, Italy; 7Medical Direction, Fondazione IRCCS Policlinico San Matteo, 27100 Pavia, Italy; 8Department of Clinical, Surgical, Diagnostics and Pediatric Sciences, University of Pavia, 27100 Pavia, Italy

**Keywords:** vaccine, SARS-CoV-2, immune response, vitamin D

## Abstract

The use of micronutrients such as vitamin D could improve the response to viral vaccines, particularly in immunosuppressed and immunosenescent subjects. Here, we analysed the association between serum 25-hydroxyvitamin D (25OHD) levels and the immune response elicited by the BNT162b2 vaccine in a cohort of 101 healthcare workers naïve for SARS-CoV-2 infection. We observed no significant differences in anti-spike (S) IgG and T-cell responses according to the 25OHD status at baseline. However, significant correlations between the 25OHD concentration at baseline and (i) the anti-S response (*p* < 0.020) and (ii) the neutralizing antibody (NT) titre (*p* = 0.040) at six months after the second dose were detected. We concluded that adequate levels of vitamin D may improve the immune response to mRNA vaccines such as BNT162b2, and that further larger studies are warranted in order to confirm these preliminary observations.

## 1. Introduction

A two-dose regimen of the BNT162b2 vaccine (Pfizer-BioNTech, Mainz, Germany) [1] was found to be safe and 95% effective against COVID-19 [2]. The BNT162b2 vaccine elicits high levels of anti-spike (S) IgG and neutralizing antibody (NT) responses in the first month after the second dose [3]. However, a significant declining trend in antibody levels was observed 70 days or more after complete vaccination [4,5]. In a prospective observational study, we showed that 98.4% of the naïve subjects vaccinated developed a positive humoral and SARS-CoV-2-specific T-cell response three weeks after the second BNT162b2 vaccine dose [6]. Moreover, a significant and rapid decrease in humoral and cell-mediated responses to the vaccine was observed within six months after vaccination [4,5,7].

Many observational studies have shown an association between a serum 25-hydroxyvitamin D (25OHD) level deficiency and the risk of developing COVID-19, its disease severity, and mortality [8,9,10,11,12]. In a large observational population study, a significant association was observed between serum 25OHD deficiency, measured before the pandemic, and the risks of SARS-CoV-2 infection and severe disease in those infected [13]. On the other hand, the combination of a multitude of variables including age, sex, comorbidity, and geographical area may affect the risk of COVID-19 infection, independent of serum 25OHD levels [14,15,16]. Furthermore, it has been demonstrated that vitamin D supplementation reduces the risk of acute respiratory infections (ARIs) [17], exerting protective effects by the expression and secretion of pro-inflammatory cytokines and chemokines [18]. These effects may be mediated in part by 25OHD reducing the ‘cytokine storm,’ which has been implicated in severe COVID-19 infection [15]. Vitamin D defends against SARS-CoV-2 infection through a complex mechanism of interactions between the innate and adaptive immune reactions [19]. A link was found between the inflammatory T helper (Th)1 program and a vitamin-D-repressed geneset. The autocrine/paracrine regulatory mechanism and its respective signals act on Th1 cell functions. The vitamin D loop enables Th1 cells to both activate and respond to vitamin D as part of a shut-down program repressing pro-inflammatory IFN-γ and enhancing anti-inflammatory IL-10 [20]. In addition, in contrast to its inhibitory effects on inflammatory cytokines, the active form of 25OHD stimulates the expression of high levels of CTLA-4, as well as FoxP3, the latter requiring the presence of IL-2 [21]. However, vitamin D has many other effects on T cells and local metabolic ‘signatures’ of skin, as demonstrated following exposure to sunlight [22]. The association between 25OHD deficiency and protection rates following vaccination is still unclear. Lee and colleagues showed no significant association between 25OHD levels and the immunogenic response to influenza vaccination. However, they observed that low 25OHD levels attenuated the immune response to strain-specific (A/H3N2 and B) influenza vaccination [23].

An analysis of the molecular mechanisms and immune signalling pathways involved suggests that high 25OHD levels may improve immune responses to different COVID-19 vaccines [19,24].

This study was conducted to evaluate the association between 25OHD serum levels and the immune response elicited by the BNT162b2 vaccine.

## 2. Methods

### 2.1. Study Design and Participants

Between 27 December 2020 and 11 February 2021, 101 healthcare workers naïve for SARS-CoV-2 infection (median age 44, IQR 34–52 years; 20 males and 81 females) were enrolled at the time of SARS-CoV-2 vaccination (at the Fondazione IRCCS Policlinico San Matteo, Pavia, Italy (Table 1). The subjects were a subgroup of those enrolled in our previous study [6].

An observational, longitudinal, and retrospective study was conducted to evaluate the association between 25OHD levels and the immune response elicited by the BNT162b2 vaccine.

Serum samples were collected at three time points: at baseline (T0, before vaccination), at three weeks (T2), and at six months (T3) after the second dose.

The study was approved by the local Ethics Committee and all subjects gave written informed consent.

### 2.2. 25OHD Levels

The serum concentration of 25OHD was determined by chemiluminescence immunoassay (Liaison, 25OH Vitamin D, Diasorin, Saluggia, VC, Italy). The lower limit of quantitation for 25OHD was 4.0 ng/mL, and the higher reportable value without dilution was 150 ng/mL.

### 2.3. Antibody Response

To evaluate the humoral response elicited by the vaccine, a quantitative characterization of Trimeric Spike (S) IgG was determined at each time point (T0, T2, and T3) by a chemiluminescent assay (Liaison SARS-CoV-2 Trimeric, Diasorin), according to the manufacturer’s instructions. The results are given as units (AU/mL) and are considered positive when ≥33.8 AU/mL.

SARS-CoV-2 NT antibodies were determined and quantified as previously reported (Percivalle E et al., 2020). Results higher than 1:10 were considered positive.

### 2.4. T-Cell Response

An ELISpot assay was used to evaluate the SARS-CoV-2 T-cell response at each time point, according to the following protocol (Cassaniti et al., CMI 2021). Briefly, peripheral blood mononuclear cells (PBMC) at a concentration of 2 × 10^5^/100 μL culture medium per well were stimulated for 24 h in 96-well plates (Merck Millipore, Darmstadt, Germany) (coated with anti-IFN-γ monoclonal capture antibody, Diaclone, France), with peptide pools (15 amino acids in length, overlapping by 10 amino acids, Pepscan, Lelystad The Netherlands) representative of the spike protein (S) at a final concentration of 0.25 µg/mL. Phytoheamagglutinin (PHA; 5 µg/mL) was used as a positive control, and medium alone as a negative control. Responses ≥10 net spot-forming cells (SFC)/million PBMC were considered positive based on background results obtained with the negative control (mean SFC + 2SD). AID ELISPOT reader system from Autoimmun Diagnostika GmbH (Strasburg, Germany) was used for count.

### 2.5. Statistical Analysis

We used the Stata software (release 17, College Station, TX, USA) for all analysis. A 2-sided *p*-value < 0.05 was considered to be statistically significant. We described continuous data with the median and 25–75th percentiles (IQR) and categorical variables as counts and percent. We computed the Spearman R to measure the association baseline 25OHD and immunity measures, together with its 95% confidence interval (CI95%) at the predefined time points. We used a multiple regression model for repeated measures to assess changes over time of the log-transformed serum 25OHD level; we computed clustered Huber–White standard errors to account for the intra-subject lack of independence. Changes over time (CI95%) were computed.

## 3. Results

### 3.1. 25OHD and SARS-CoV-2 Antibody Response

Serum 25OHD concentration was tested at baseline, before vaccination (T0; median concentration: 16.20 ng/mL; IQR 11.50–22.90), after the complete course of vaccination (T2; median concentration: 14.55 ng/mL IQR 10.30–19.50), and at six months after the second vaccine dose (T3; median concentration: 26.25 ng/mL IQR 19.65–32.80) (Figure 1A).

At T2, all subjects developed a positive anti-trimeric-spike (S) response (median level: 800.0 AU/mL; IQR 692–800.0) over the limit of the quantifiable range of the assay. However, we observed a significant decrease in the anti-S response (median level: 176 AU/mL; IQR 119.50–301; *p* < 0.001) at T3 (Figure 1B). No significant correlation was observed between baseline 25OHD levels and the anti-S response at T2 (Spearman R = −0.105; CI95% −0.299–0.096; *p* < 0.304), but a significant correlation at T3 (Spearman R = 0.233; CI95% 0.037–0.412; *p* < 0.020) was detected (Figure 2A,B).

A similar trend was observed in NT antibodies: at T2, serum NT titres (median NT titre: 320; IQR 320–640) reached significantly higher levels (*p* < 0.001) compared to T3 (median NT titre: 40; IQR 20–40) (Figure 1C). At T2, no significant correlation was observed between baseline 25OHD levels and NT titres (Spearman R = −0.075; CI95% −0.268–0.124; *p* = 0.460), while at T3, the 25OHD concentration and NT titres correlated significantly, though weakly (Spearman R = 0.206; CI95% 0.009–0.388; *p* = 0.040) (Figure 2C,D).

### 3.2. 25OHD and SARS-CoV-2 Specific T-Cell Response

The S-specific T-cell response was determined and associated to the baseline serum 25OHD concentration.

All subjects developed a detectable response at T2 (median response 115 SFC/million PBMCs; IQR 65–205), which was significantly higher (*p* < 0.001) compared to T3 (Figure 1D).

There was no significant association between the baseline 25OHD concentration and S-specific T-cell response at T2 (Spearman R = 0.024; CI95% −0.175–0.222; *p* = 0.808), nor at T3 (Spearman R = 0.002; CI95% −0.200–0.204; *p* = 0.984) (Figure 3A,B).

### 3.3. Changes of 25OHD and Immune Response

The serum 25OHD concentration increased significantly at T3 compared to T0 and T2 (regression model for repeated measures, change in log serum 25OHD 0.39, CI95% 0.33–0.45; *p* = 0.001), but no association was observed between the variation in 25OHD levels and age or gender. No significant correlation was observed between the variation in 25OHD levels over time, considered as the difference between T2 and T0 and the anti-S response at T2 (Spearman R = −0.023; CI95% −0.223–0.183; *p* = 0.828). No significant correlation was observed between the variation in 25OHD levels (the difference between T3 and T2) and the anti-S response at T3 (Spearman R = 0.112; CI95% −0.093–0.309; *p* = 0.282). However, the data showed that the changes in 25OHD levels correlated, though weakly, with NT titres at T2 (Spearman R = 0.298; CI95% 0.104–0.471; *p* = 0.003), but not at T3 (Spearman R = 0.014; CI95% −0.191–0.218; *p* = 0.894).

No significant association was observed between changes in 25OHD and S-specific T-cell responses (T2: Spearman R =−0.046; CI95% −0.248–0.158; *p* = 0.655, and T3: Spearman R =−0.101; CI95%−0.304–0.109; *p* = 0.343).

## 4. Discussion

We evaluated: (i) the correlation between 25OHD serum levels before vaccination and the immune response elicited by the BNT162b2 vaccine and (ii) the changes of 25OHD levels up to six months after vaccination and the association with the trend of immunity.

In our study, we reported a weak but significant correlation between 25OHD concentration at baseline and anti-S IgG levels and NT titres six months after complete vaccination. This suggests that 25OHD levels at the time of vaccination may impact the persistence of the antibody response. Conversely, we did not find a significant association between baseline 25OHD levels and the antibody response observed 21 days after vaccination. However, since most vaccinated subjects showed anti-S IgG and NT titres at the upper limit of the quantifiable ranges of the assays adopted, we cannot exclude that this may have impacted the correlation analysis and that a more precise determination of the antibody levels could have provided different results.

Serum neutralizing antibodies have been shown to correlate with protection following SARS-CoV-2 natural infection or vaccination [25,26,27].

However, many studies have shown that the efficacy of vaccination decreases over time and so does immunity, requiring booster doses of vaccine [28,29]. It is tempting to speculate that the duration of vaccine effectiveness may be longer in individuals with higher 25OHD levels. The vaccine-induced immune response appears to be weaker in older people, including lower concentrations of neutralizing antibodies [30,31]. Furthermore, the vaccination may not be as efficacious in certain vulnerable populations, such as the elderly and autoimmune disease or transplant patients. The potential impact of 25OHD supplementation on vaccine effectiveness in these patient categories could be evaluated.

In contrast to what was observed for the antibody response, we did not observe a correlation between 25OHD levels and the number of S-specific T-cells producing IFNγ. However, we cannot exclude that other T-cell functions besides IFNγ production may be affected by 25OHD levels. Vitamin D has known effects on innate and adaptive immunity. Multiple pleiotropic effects have been demonstrated on the actions of vitamin D at the anti-inflammatory and immunomodulatory levels [32,33]. Vitamin D affects the maturation of T cells with a decrease in the inflammatory Th17 phenotype and an increase in T regulatory cells.

A direct effect on CD4^+^ T cells has been shown to enhance the development of Th2 cells by decreasing the production of inflammatory cytokines (IL-17, IL-21) and increasing the production of anti-inflammatory cytokines, such as IL-10 [34,35].

The vaccine-induced immune response activates CD4+ T cells with a T helper (THαβ)-type cytokine bias and CD8+ T cells with a cytotoxic phenotype. Vitamin D promotes the association of THαβ cells with anti-virus immunity, which improves the production of interleukin-10, antiviral IFN-I, and IgG1 from B-cell lineages [36,37]. Adequate levels of 25OHD can aid THαβ-type immunity and promote the activation of B cells with higher levels of IgG neutralizing antibodies [19].

A recent study showed that the vaccination response, measured as SARS-CoV-2 IgG concentration, does not depend on 25OHD status in healthy adults [38], as reported in prior studies on the effect of 25OHD concentrations and the immunogenicity of influenza vaccination in elderly people [23,39].

However, in our analysis, we have shown an association between serum 25OHD levels and both anti-S IgG and neutralizing antibodies after vaccination.

An increase in 25OHD serum levels compared to baseline was observed at six months after vaccination. This can be explained by the fact that the six months coincided with the summer season, therefore increased vitamin D can be obtained by exposure to the sun.

A limitation of our study resides in the analysis of the lack of older individuals. Therefore, the study of the correlation of 25OHD and vaccine-induced immune responses should be expanded to vulnerable populations.

In conclusion, the results of this study show that baseline 25OHD levels are correlated with the persistence of antibody levels after administration of an mRNA-based vaccine. Since the adoption of this vaccine platform is likely to expand in the near future, 25OHD supplementation before or at the time of vaccination may be considered as a potential option to increase vaccine responsiveness.

However, further larger studies are warranted in order to confirm these preliminary observations, particularly in fragile populations.

## Figures and Tables

**Figure 1 biomedicines-10-01993-f001:**
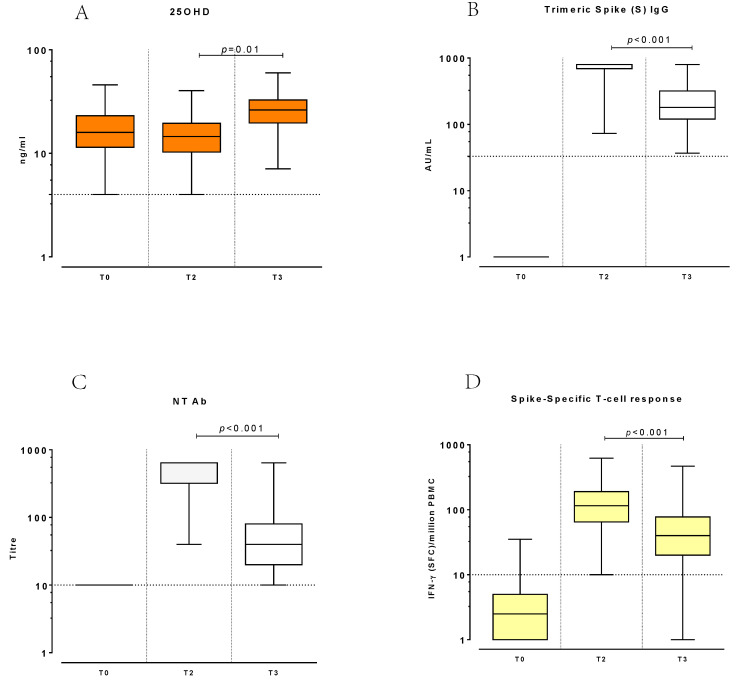
Trend of 25OHD levels (**A**), Trimeric Spike (S) IgG response (**B**), and anti-SARS-CoV-2 neutralizing (NT) antibodies (**C**), and T-cell responses (**D**) measured at T0, T2, and T3.

**Figure 2 biomedicines-10-01993-f002:**
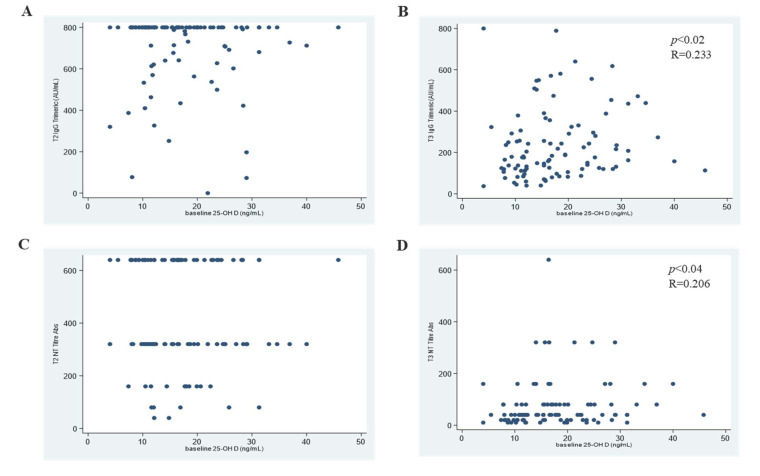
Correlation between 25OHD concentration at baseline and anti-S response at T2 (**A**) and T3 (**B**); 25OHD concentration vs. NT titre at T2 (**C**) and T3 (**D**).

**Figure 3 biomedicines-10-01993-f003:**
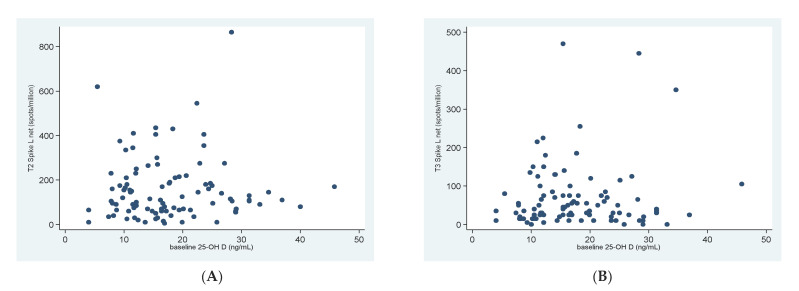
Correlation between 25OHD concentration at baseline and T-cell response at T2 (**A**) and T3 (**B**).

**Table 1 biomedicines-10-01993-t001:** Subjects’ characteristics.

GENDER			
Male (%)	20 (19.8)		
Female (%)	81(80.2)		
Median Age (IQR)	44 (34–52)		
TIME POINTS	T0	T2 (21 days; 19–44)	T3 (161 days; 110–276)
25OHD (ng/mL)	16.2; 11.5–22.9	14.55; 10.3–19.5	26.25; 19.65–32.8
Anti-Trimeric Antibody (AU/mL)	N.A	800.0; 692–800.0	176.0; 119.5–301
Serum neutralizing titer	N.A	320; 320–640	40; 20–80
Anti-Spike T cell response (SFC/million PBMCs)	N.A	115; 65–205	40; 20–77.5
(Median, IQR)			

## Data Availability

Data is contained within the article.

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
