# Peer review of "Association between Vitamin D Serum Levels and Immune Response to the BNT162b2 Vaccine for SARS-CoV-2"

_biomedicines, 2022, doi:10.3390/biomedicines10081993_

Round 1
Reviewer 1 Report
This study investigates whether vitamin D levels may affect the efficacy of vaccination with the BNT162b2 vaccine against SARS-CoV-2. In a cohort of 101 health care workers the authors observed that vitamin D levels did not affect the development of immune responses at 21 days after infection. However they found slight and statistically significant effects at 6 months after vaccination with two doses. This is a well performed study suggesting that vitamin D levels might indeed positively affect the development of a protective immune response. One drawback, however, is that the study cohort was quite small and thus, further studies will be necessary to confirm this finding.
Author Response
Thank you for considering the our manuscript.
Reviewer 2 Report
The authors tested association between levels of 25OHD, the functional derivative of vitamin D, and correlates of humoral protection from the Pfizer COVID-19 vaccine. The sample size (101) is quite small. The authors observed weak, but statistically significant correlations six months post vaccination but not at two earlier time points.
Some things to correct:
1. The explanation of the known roles of vitamin D in immunity at lines 47-53 and lines 184-191 is inadequate. The authors should look at these three references and incorporate their contents:
Chauss D, et al. Autocrine vitamin D signaling switches off pro-inflammatory programs of TH1 cells. Nature Immunology 2022; 23:62-74.
Sigmundsdottir H, et al. DCs metabolize sunlight-induced vitamin D3 to "program" T cell attraction to the epidermal chemokine CCL27. Nat Immunol. 2007;8:285-293.
Jeffery LE, et al. 1,25-Dihydroxyvitamin D3 and IL-2 Combine to Inhibit T Cell Production of Inflammatory Cytokines and Promote Development of Regulatory T Cells Expressing CTLA-4 and FoxP3. J Immunol. 2009;183:5458
2. The set of references to justify studying vitamin D in conjunction with protection from COVID-19 (at lines 44-45) is inadequate. At least the following seven additional references should be cited and there are many more such studies that could be cited.
Maghbooli, Z et al., Vitamin D sufficiency, a serum 25-hydroxyvitamin D at least 30 ng/mL reduced risk for adverse clinical outcomes in patients with COVID-19 infection. PLoS One 2020;15(9):e0239799.
Merzon E, et al. Low plasma 25(OH) vitamin D level is associated with increased risk of COVID-19 infection: an Israeli population-based study. FEBS J. 2020; 13:644
Panagiotou G. et al. Low serum 25-hydroxyvitamin D (25[OH]D) levels in patients hospitalised with COVID-19 are associated with greater disease severity. Clin Endocrinol. 2020; 93:508-511.
Li Y, Tong CH, Bare LA, Devlin JJ Assessment of the association of vitamin D level with SARS-CoV-2 seropositivity among working-age adults. JAMA Netw Open 2021; 4(5):e2111634
Teshome A, Adane A, Girma B, Mekonnen ZA. The impact of vitamin D level on COVID-19 infection: systematic review and meta-analysis. Front Public Health 2021; 9:624559
Wang Z, Joshi A, Leopold K, Jackson S, Christensen S, Nayfeh T, Mohammed K, Creo A, Tebben P, Kumar S. Association of vitamin D deficiency with COVID-19 infection severity: systematic review and meta-analysis. Clin Endocrinol. 2022; 96:281-287.
Israel A, et al. Vitamin D deficiency is associated with higher risks for SARS-CoV-2 infection and COVID-19 severity: a retrospective case-control study. Intern Emerg Med. 2022 Jun;17(4):1053-1063.
3. Lines 25-27, Change,
"However, a significant correlation between 25OHD concentration at baseline and 25 anti-S response (p<0.020) at six months after the second dose and neutralizing antibodies (NT) titre (p=0.040) was detected."
to
"However, significant correlations between 25OHD concentration at baseline and
i) anti-S response (p<0.020) and
ii) neutralizing antibodies (NT) titre (p=0.040) at six months after the second dose
were detected."
The original phrasing has multiple grammar errors.
4. At lines 104-105, the authors wrote: "we computed clustered Huber-White standard errors to account for intra-subject lack of independence", but I do not see where such standard errors are shown or used subsequently.
5. In Table 1, the headings T2 (21; 19-44) T3 (161; 110-276) do not make sense
6. The time point definitions are confusing. In Table 1, the authors use T0, 72, T3. However, in the Figure 1 legend, the authors use T1, T2, T3. I cannot find. see where in the manuscript T1 is defined.
7. Line 181, change "Conversely to what observed" to "In contrast to what we observed" [grammar error]
8. Lines 216-217, Change "this preliminary observations, particularly in fragile populations." to "these preliminary observations, particularly in fragile populations" [grammar error]
9. The references are misformatted with the numbers appearing twice
